# Research on Path Planning of Mobile Robot Based on Improved Theta* Algorithm

Yi Zhang [1], Yunchuan Hu [1,*], Jiakai Lu [2] and Zhiqiang Shi [3]

1 School of Advanced Manufacturing Engineering, Chongqing University of Posts and Telecommunications, Chongqing 400065, China
2 Key Laboratory of Optoelectronic Information Sensing and Technology, Chongqing University of Posts and Telecommunications, Chongqing 400065, China
3 China Assistive Devices and Technology Center, Beijing 100050, China
* Correspondence: s202101007@stu.cqupt.edu.cn

**Abstract:** The Theta* algorithm is a path planning algorithm based on graph search, which gives the optimal path with more flexibility than A* algorithm in terms of routes. The traditional Theta* algorithm is difficult to take into account with the global and details in path planning and traverses more nodes, which leads to a large amount of computation and is not suitable for path planning in large scenarios directly by the Theta* algorithm. To address this problem, this paper proposes an improved Theta* algorithm, namely the W-Theta* algorithm. The heuristic function of Theta* is improved by introducing a weighting strategy, while the default Euclidean distance calculation formula of Theta* is changed to a diagonal distance calculation formula, which finally achieves a reduction in computation time while ensuring a shorter global path; the trajectory optimization is achieved by curve fitting of the generated path points to make the motion trajectory of the mobile robot smoother. Simulation results show that the improved algorithm can quickly plan paths in large scenarios. Compared with other path planning algorithms, the algorithm has better performance in terms of time and computational cost. In different scenarios, the W-Theta* algorithm reduces the computation time of path planning by 81.65% compared with the Theta* algorithm and 79.59% compared with the A* algorithm; the W-Theta* algorithm reduces the memory occupation during computation by 44.31% compared with the Theta* algorithm and 29.33% compared with the A* algorithm.

**Keywords:** Theta* algorithm; A* algorithm; weighting strategy; path planning; mobile robots; trajectory optimization





## 1. Introduction

### 1.1. Research Background

With the development of robotics and artificial intelligence, the application of robots can be seen in all walks of life. The most important manifestation of mobile robot intelligence is autonomous navigation, which is mainly to solve the problem of "where am I", "where should I go" and "how should I go" of robots. The key technology to solve this problem is path planning, which is also a difficult technology. Path planning requires the robot to find the best collision-free path from the starting point to the target point in different map environments, and the quality of the path planning can directly affect the robot's task completion.

Nowadays, the rise of mobile robots, unmanned vehicles, and the field of UAVs has led more and more scholars to work on path planning algorithms. Although there are many algorithms proposed for path planning, there are still great challenges in the research of path planning, especially about the trajectory optimization of paths, and there are still many problems that need to be solved. The paths planned by the current mainstream path

planning algorithms are shown in Figure 1, where path A and path B are connected by discrete path points, so there are problems such as paths turning at acute angles and the planned paths are not the actual shortest paths, which are not ideal for the movement of mobile robots. In the actual application of mobile robots, it is necessary to avoid sudden sharp turns or stops, and to ensure the continuity of posture, speed and acceleration at the turns during the movement. For example, in unmanned driving, the discontinuity of the planned path interruptions leads to sudden turns and sudden changes in speed and acceleration during the driving process, which can affect the safety of passengers. Secondly, the path planning based on grid map will restrict the robot to search the direction on the map, so that the planned path is not the actual shortest path, which will generate many useless inflection points, and these inflection points will affect the operation efficiency of the robot. As shown in Figure 1 for path C, a shorter, smoother path represents less energy loss and higher efficiency while ensuring the safety of the mobile robot. For example, in the process of transporting goods by logistics AGVs, the efficiency of the AGVs directly affects the overall efficiency of the logistics warehouse operation.

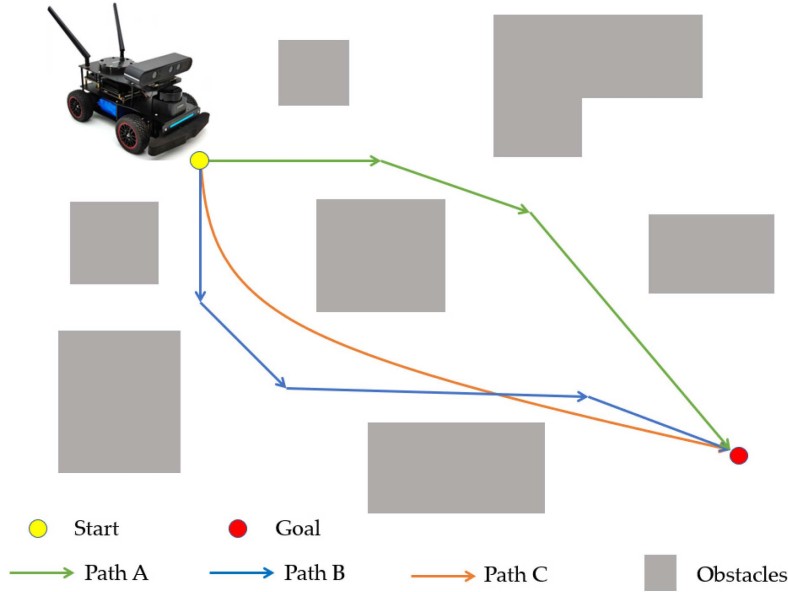

**Figure 1.** Path planning diagram.

To address the above problems, this paper will propose a solution by adding multi-angle pathfinding idea and path smoothing technique to the existing path planning algorithm. Multi-angle pathfinding is to break the constraints of the traditional path planning algorithm in the search direction brought by the discrete grid map, so that its path can be extended from any angle. The path smoothing process will make it easier to realize the continuous change of position transformation, moving speed, and acceleration of the robot in the process of motion.

### 1.2. Related Work

The current mainstream path planning algorithms are broadly classified into three main categories: population intelligence-based algorithms, random sampling-based, and graph search-based. The idea of cluster-based optimization is inspired by the process of population evolution, and its typical algorithms include the genetic algorithm proposed by J. Holland [1,2], the ant colony algorithm proposed by Marco Dorigo [3,4], and the particle swarm algorithm proposed by Eberhart and Kennedy [5], etc. The random sampling-based path planning method gradually obtains the connectivity information between different states in the state space by random sampling and then gradually finds the feasible solution, which greatly reduces the search complexity and improves the search efficiency, but these methods have the problem of poor stability of the solution quality and cannot guarantee

the optimal solution. The most studied methods are the Probabilistic Roadmap (PRM) [6] algorithm and the Rapidly expanding Random Tree (RRT) algorithm [7]. The graph search-based path planning method first requires the construction of graphs, i.e., the state space of the mobile robot system is partitioned into discrete spaces (considered as graphs) according to a certain criterion, after which graph search algorithms are applied to search for paths in these graphs, and the commonly used algorithms are Dijkstra's algorithm [8,9], A* algorithm [10,11], and Theta* algorithm [12–15].

A* algorithm based on graph search technique has a wide range of applications in path planning for robots. The advantage of A* algorithm is that it is better in real time and easy to implement, but the disadvantage is that the uncertainty of the heuristic function affects the quality of path planning [16], and the search direction of A* algorithm is limited by the shape of the grid in the map in the process of path planning, so the planned path is not the shortest path in the actual map shortest path in the actual map. To address this problem, Alex Nash et al. proposed the Theta* algorithm [14]. This algorithm is an improved branch based on the A* algorithm, which combines the respective features of the A* algorithm and the visual map method, and searches along the grid on the grid map during path planning, but the path is not restricted to the edges of the grid, which achieves the arbitrary turning angle and effectively reduces the length of the path and the number of turning points. The difference between the Theta* algorithm and the A* algorithm is that the Theta* algorithm allows the parent of the current node to be any other node, while in the A* algorithm, the parent of the current node can only be a visible neighbor of the node. Since the Theta* algorithm is derived from the A* algorithm, it is still limited in operational efficiency by the excessive number of nodes traversed in the path search process.

The optimal path proposed by path planning is not only the shortest passable path, but also the continuous and smooth transition of the robot's position, velocity, and acceleration during the movement, which requires the optimization of the planned path. Many scholars have proposed different path optimization schemes in previous studies. The most commonly used methods for path smoothing optimization are Bézier curves [17,18] and B-sample curve optimization [19]. The Bézier curve approach allows curve fitting of the generated path points during path planning and thus achieves the role of smoothing the path. Usually, the number of initial path points obtained by the global path planning algorithm is small, resulting in a low fit, and therefore the smoothed path may collide with the obstacle. In order to increase the curve fit, the path points are usually increased. However, if the number of path points is increased excessively, it increases the computational effort of the algorithm thus leading to a decrease in the efficiency of the algorithm [20]. Another aspect of spline curve fitting is only for the geometric smoothing of the path, and the continuity of the robot's running speed and acceleration is not taken into account. Qinming Hu et al. [21] proposed a polynomial interpolation-based method to improve the continuity problem of the robot during navigation. Polynomial interpolation [22] is a simple functional method. This method is solved by finding a polynomial containing all path nodes and adding continuity constraints for node position, velocity, and acceleration.

To address the problems of low computational efficiency and long convergence time of Theta* algorithm, this paper proposes a W-Theta* algorithm, which is based on the dynamic weighted improvement of Theta* algorithm to improve the path planning efficiency of Theta* in large scenarios, while using the polynomial optimization method based on minimum jerk [23,24], so that the W-Theta* algorithm to satisfy smoothness and continuity of the planned paths. Firstly, a two-dimensional grid map is constructed to provide a simulation environment for validating the algorithm; secondly, the heuristic function of Theta* is improved, while the diagonal distance expression is used, which finally achieves the reduction of computation time while ensuring the shortest global path; finally, the generated path points are curve-fitted to make the robot's motion trajectory smoother.

## 2. Algorithm Design

### 2.1. Theta* Algorithm

The Theta* algorithm is a variant of the A* algorithm, which is centered on the grid of the current node and expands outward, the node expansion direction of the A* algorithm is shown in Figure 2 and is restricted to eight directions around the current node, while the same Theta* does not restrict the expansion direction of the node. The Theta* algorithm, like the A* algorithm, is applicable to static grid maps and its cost function expression is:

$$f(n) = g(n) + h(n) \tag{1}$$

where $f(n)$ is the estimated cost from the starting point to the target point; $g(n)$ is the actual cost from the starting point to the current node $n$; and $h(n)$ is the heuristic function, which represents the estimated cost from the intermediate point $n$ to the target point. In the traditional Theta* algorithm, the calculated value of $g(n)$ is generally characterized as the number of nodes of the grid map through which the path passes, and the heuristic function $h(n)$ is the distance from node $n$ to the target point.

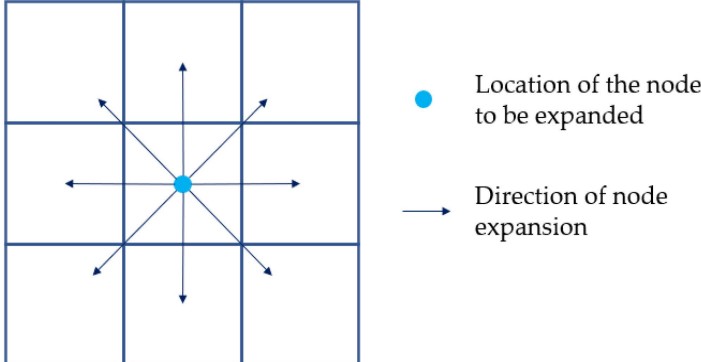

**Figure 2.** A* algorithm node search direction.

The key difference between the Theta* algorithm and the A* algorithm is that the parent node of a vertex in the A* algorithm can only be the node adjacent to it, while the visibility check mechanism is added in the Theta* algorithm to break the restriction of the raster environment, so that the parent node of a node can be any node. When extending a new node, the A* algorithm considers only one path, while the Theta* algorithm considers two paths, as shown in Figure 3a, when the Theta* algorithm wants to extend from point A2 to point C3.

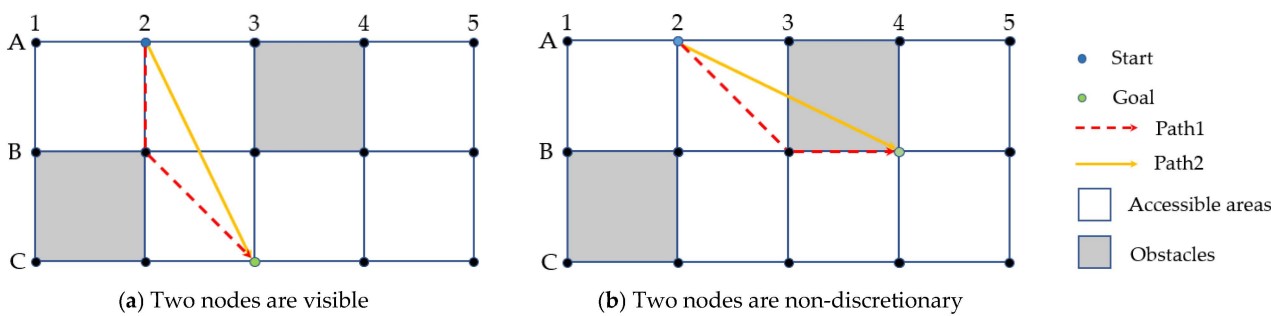

(**a**) Two nodes are visible      (**b**) Two nodes are non-discretionary

**Figure 3.** Theta* algorithm path planning schematic.

1.  Path 1 has 3 path nodes: A2, B2, and C3, currently the parent node of C3 is B2 and the parent node of B2 is A2. This is also the path considered by the A* algorithm.
2.  Path 2 has 2 path nodes: A2 and C3. Currently, C3's parent node is A2. This is an additional path to consider with the A* algorithm.

According to the triangle trilateral length property it is known that the distance of path 2 must be shorter than path 1. Additionally, the need to visualize the current node and the extended node to determine whether there are obstacles before the two nodes, as in Figure 3a, there is no obstacle between node A2 and node C3, so path 2 can be selected, but in Figure 3b, there is an obstacle between node A2 and node B4, the two nodes are not visible, so path 2 cannot be selected.

The specific steps of the Theta* algorithm are as follows:

1.  Initialize the starting point and target point, create a new open table and a new close table, and add the starting point to the open table.
2.  Determine whether the open table is empty, if it is empty, the path planning fails and the search stops; if it is not empty, the node with the lowest evaluation cost in the open table is taken as the current node to be expanded and set to $n$. Go to the next step.
3.  Determine whether the current node $n$ is the target point, if yes, it means the path finding is successful, backtrack the parent node of n until the starting point is the path, and the search is terminated; otherwise, go to the next step.
4.  Extend node $n$. Iterate over all neighboring nodes of node $n$ (set to $s$) and perform the following operations.
    *   If node $s$ is an obstacle or is already in the close table, do not process it; instead proceed to the next step.
    *   If node $s$ is not in the open list, initialize the actual cost $g(s)$ from the starting point to node $s$ to infinity, set its parent node to NULL, and insert node $s$ into the open table. If node $s$ is in the open list, proceed directly to the next step.
    *   Update the information of the node. Check if the parent node (set to $p$) of the current node $n$ exists. If it exists, check whether node $s$ and node $p$ are visible by line of sight (LOS) function [13]. If the two nodes are visible and the $g(p)$ of node $p$ plus the generation value from node $p$ to node $s$ is less than the $g(s)$ of node $s$, the parent of node $s$ is updated to node $p$, and the $g(s)$ of node $s$ is updated to the $g(p)$ of node $p$, plus the generation value from node $p$ to node $s$. If the two nodes are not visible and the $g(n)$ of node $n$ plus the cost from node $n$ to node $s$ is less than the $g(s)$, then the parent node of node $s$ is updated to node $n$ and $g(s)$ of node $s$ is updated to $g(n)$ of node $n$ plus the generation value from node $n$ to node $s$.
    *   The original $g(s)$ of node $s$ is compared with the updated actual generation value, and if the updated actual cost is smaller than the original actual cost, the estimated cost of node $s$ is recalculated and the information related to node $s$ in the open table is updated.
5.  Remove the already traversed node $n$ from the open table and add it to the close table, then return to step 2.

Based on the description of the steps of the Theta* algorithm above, its pseudo-code can be written as shown in Algorithm 1.

---

**Algorithm 1.** Theta* algorithm.

---

**Input:** start point:{$s_{start}$}, map arrays:{map}, goal point:{$s_{goal}$}
**Output:** algorithm planning path: {path}
  1:   Initialize the open and close tables : {open := closed := ∅}
  2:   Initializing the cost function: {$g(s_{start})$ :=0}
  3:   Set the start node as the parent node: {parent($s_{start}$) :=$s_{start}$}
  4:   open.Insert($s_{start}$, $g(s_{start})$+$h(s_{start})$)
  5:  **while** open ≠ ∅ **do**
  6:     s :=open.Pop()
  7:    **if** s = $s_{goal}$ **then**
  8:       **return** path
  9:    **end if**
 10:    closed := closed := ∪ {s}
 11:    **for** s′ ∈ nghbr$_{vis}$(s) **do**
 12:      **if** s′ ∉ closed **then**
 13:        **if** s′ ∉ open **then**
 14:          g(s′) := ∞
 15:          parent(s′) :=NULL
 16:        **end if**
 17:        $g_{old}$ :=g(s′)
 18:        **if** lineofsight(parent(s), s′) **then**
 19:          **if** g(parent(s)) + cost(parent(s), s′) < g(s′) **then**
 20:            g(parent(s′)) :=parent(s)
 21:          g(s′) :=g(parent(s)) + cost(parent(s), s′)
 22:          **end if**
 23:        **else**
 24:          **if** g(s) + c(s, s′) < g(s′) **then**
 25:           parent(s′) :=s
 26:           g(s′) :=g(s) + cost(s, s′)
 27:          **end if**
 28:        **end if**
 29:        **if** g(s′) < $g_{old}$ **then**
 30:          **if** s′ ∈ open **then**
 31:            open.Remove(s′)
 32:          **end if**
 33:          open.Insert(s′, g(s′) + h(s′))
 34:        **end if**
 35:      **end if**
 36:    **end for**
 37:  **end while**

---

### 2.2. Improvement of Heuristic Function

The heuristic function $h(n)$ of Theta* is mainly to estimate the cost of the robot's current position node to the target point, and different $h(n)$ will have different effects on the operation efficiency of the algorithm. When the value of $h(n)$ is 0, the Theta* algorithm will degenerate into Dijkstra algorithm, which can guarantee that the output path is optimal, but the number of nodes diffused in the process of calculation is large, resulting in low efficiency in path planning in a large scene environment and cannot meet the demand of real-time; when the value of $h(n)$ is very large, Theta* can quickly plan at a path, but cannot guarantee that the path is the shortest path, so it defeats the purpose of the path planning algorithm. Therefore, in order to ensure that the path planning algorithm meets the two conditions of shortest path and low computational cost, it is necessary to choose a suitable heuristic function for the algorithm.

The heuristic function of the improved algorithm is mainly to find the shortest path. Using the traditional method of expanding nodes will generate many useless expansion nodes and increase the computational cost. To address this problem, the following improvements are made in this paper:

1. To get the optimal path, the predicted cost calculated by the heuristic function must be less than or equal to the actual minimum cost and the closer the two are, the more efficient the search is. In addition, the cost between two points in two-dimensional space usually refers to the Euclidean distance between the two, so this paper uses the Euclidean distance to express the distance between the current node and the end point, and its formula can be expressed as follows:

$$d(n) = \sqrt{(x_n - x_{goal})^2 + (y_n - y_{goal})^2} \tag{2}$$

    where $(x_n, y_n)$ are the coordinates of the current node $n$ and $(x_{goal}, y_{goal})$ are the coordinates of the target point.

2. If the cost function $f(n)$ of the current node $n$ corresponds to more than one path, all of these paths will be searched, but only one of them is actually needed. This situation occurs very frequently in maps with few obstacles. To solve this problem, in this paper, we will add additional values to the heuristic function $h(n)$, the size of which is the vector fork product of the initial point-to-target vector and the current point-to-target vector, and then change the value of $h(n)$ to make it more inclined to the connection from the initial point to the target point in selecting the path, to ensure uniqueness in planning the path and reduce unnecessary computation. The functional expression is:

$$c = V_{c\_e} \times V_{s\_e} \tag{3}$$

    where c denotes the vector fork product of the start-point-to-target-point vector $V_{s\_e}$ and the current node-to-target-point vector $V_{c\_e}$.

3. By adding a weight $w$ to the heuristic function $h(n)$ and then dynamically adjusting this weight according to the progress of the algorithm, the importance of the heuristic function is reduced by decreasing the weight as the path planned by the algorithm approaches the target point, while increasing the relative importance of the true cost of the path. According to the above three improvements, the expression of the heuristic function is:

$$h(n) = w \left[ d(n) + \frac{c}{d(s)} \right] \tag{4}$$

    where $w$ is the weight and its size is set to $w = 1 + \frac{d(n)}{d(s)}$; $c$ is the crossover operator that breaks the path balance; $d(n)$ is the distance from the current node $n$ to the target point, expressed using the Euclidean distance; $d(s)$ is the Euclidean distance from the starting point to the target point.

### 2.3. Trajectory Optimization

The output of the Theta* algorithm is a global optimal path point based on a 2D grid map, and these path points may be sparse and unsmooth, which can lead to problems such as spinning in place and poor movement of the robot during motion. Therefore, the generated paths need to be optimized so that the robot can achieve smooth steering during operation.

The three objectives of path length, steering smoothness and steering safety need to be considered when smoothing the path. The optimization method proposed in this paper represents the trajectory by nth order polynomial as shown in Equation (6).

$$f(t) = p_0 + p_1 t + p_2 t^2 + \ldots + p_n t^n = \sum_{i=0}^{n} p_i t^i \tag{5}$$

where $p_0 \sim p_n$ are the trajectory parameters, which can be set as a parameter vector as shown in Equation (7).

$$p = [p_0, p_1, \ldots, p_n]^T \tag{6}$$

Thus, with Equations (6) and (7), the trajectory function f can be expressed in vector form as:

$$f(t) = [1, t, t^2, \ldots, t^n] \cdot p \tag{7}$$

By deriving the trajectory function, the position, velocity, acceleration, jerk, snap, etc. of the trajectory at any moment can be obtained, and the specific function expressions are as follows.

$$v(t) = f'(t) = [0, 1, 2t, 3t^2, 4t^3, \ldots, nt^{n-1}] \cdot p \tag{8}$$

$$a(t) = f''(t) = [0, 0, 2, 6t, 12t^2, \ldots, n(n-1)t^{n-2}] \cdot p \tag{9}$$

$$jerk(t) = f^{(3)}(t) = [0, 0, 0, 6, 24t, \ldots, \frac{n!}{(n-3!)}t^{n-3}] \cdot p \tag{10}$$

$$snap(t) = f^{(4)}(t) = [0, 0, 0, 0, 24, \ldots, \frac{n!}{(n-4!)}t^{n-4}] \cdot p \tag{11}$$

Since a complex trajectory cannot be represented by a polynomial, it is necessary to divide the trajectory into multiple local trajectories in time, and then represent the local trajectories by a polynomial each.

$$f(t) = \begin{cases} [1, t, t^2, \ldots, t^n]p_1 & t_0 \le t < t_1 \\ [1, t, t^2, \ldots, t^n]p_2 & t_1 \le t < t_2 \\ \quad \ldots \ldots \\ [1, t, t^2, \ldots, t^n]p_k & t_{k-1} \le t < t_k \end{cases} \tag{12}$$

where $k$ is the number of segments of the trajectory and $p_i$ is the parameter vector of the $i$th segment of the trajectory.

The objective function of the minimum jerk is to solve the parameter vector of each trajectory to minimize the value of the jerk function, and it also needs to satisfy the constraints. The objective function of minimum jerk can be obtained from the above:

$$\min(jerk(t)) = \min \int_0^T \left(f^{(3)}(t)\right)^2 dt = \min \sum_{i=1}^k \int_{t_{i-1}}^{t_i} \left(f^{(3)}(t)\right)^2 dt \tag{13}$$

According to the optimization function to add constraints, there are mainly two kinds of constraints without considering obstacles, one is equation constraint, which mainly constrains the initial state and termination state of the trajectory, as well as the start and end position of each section of the trajectory; the other is continuity constraint, which can make the adjacent trajectories smoothly transition. The optimized effect is shown in Figure 4.

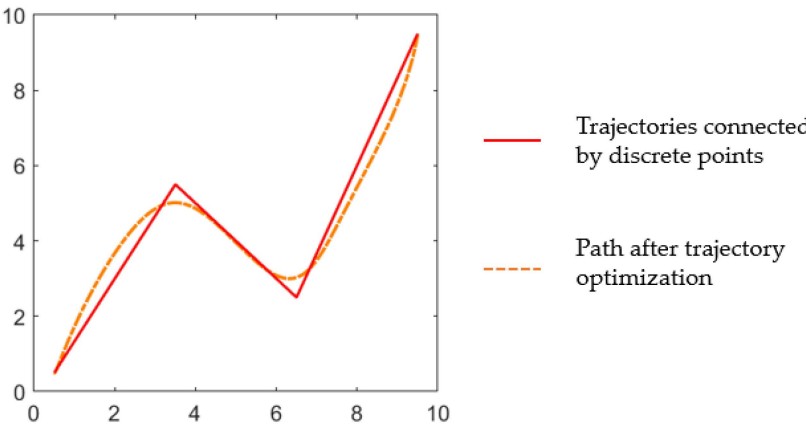

**Figure 4.** Trajectory optimization effect figure.

## 3. Simulation

In order to verify the feasibility and search efficiency of the algorithm proposed in this paper, this validation experiment is divided into two parts. The first part mainly tests the effectiveness and stability of the W-Theta* algorithm in an environment with randomly generated obstacles of different densities under maps of different sizes. The second part mainly compares the W-Theta* algorithm with the original Theta* algorithm, A* algorithm and Dijkstra algorithm mainstream path algorithms in different scenarios, respectively. The second part of the experiments compares the time cost, the number of path nodes traversed, the path length, and the search computation cost of the algorithms using the average data of 10 experiments as the experimental results. The experiments were conducted on a computer with Intel Core i7-10875H, 2.3 GHz CPU, and 16G RAM, and the algorithm experimental platform used Matlab software, whose version number is 2022a.

### 3.1. W-Theta* Algorithm Stability Test Experiment

This part of the experiment was designed with 3 groups of maps, each with dimensions of $40 \times 40$ m, $80 \times 80$ m, and $100 \times 100$ m, respectively, where each group of the same size map has an obstacle coverage of 10, 30, and 50%. As shown in Figure 5, the green point is the starting point, the red point is the ending point, and the black grid is the obstacle. The solid blue line is the path planned by Theta* algorithm, and the dashed orange line is the path planned by W-Theta* algorithm.

The data of the W-Theta* algorithm and Theta* algorithm run on each map are shown in Table 1. The data in the table shows that the pathfinding time of the W-Theta* algorithm is shorter than that of the Theta* algorithm for maps of different sizes and obstacle densities, and the W-Theta* algorithm reduces the pathfinding time compared with the Theta* algorithm for maps of $40 \times 40$ m, $80 \times 80$ m, and $120 \times 120$ m by 47.12%, 84.77%, and 92.56%, respectively. The difference is more obvious for larger maps. Usually, the time spent by an algorithm can visually reflect the time complexity of the algorithm, and it can be concluded that the time complexity of the W-Theta* algorithm is less than that of the original Theta* algorithm. In this paper, the heuristic function was optimized to achieve the control algorithm in planning the path to traverse the path nodes. This reduced the computation time and computation cost of the algorithm. Through this experiment, it can be seen that the W-Theta* algorithm calculates a different number of traversed map nodes than the Theta* algorithm; in the same size of the map, the greater the density of obstacles, the greater the gap between the two in the map size of the larger the gap. The larger the map size, the greater the difference between the W-Theta* algorithm and Theta* algorithm. In the maps of $40 \times 40$ m, $80 \times 80$ m and $120 \times 120$ m, respectively, the number of traversed nodes is reduced by 36.32, 72.58, and 82.83%. W-Theta* in these three groups of experiments calculates the number of traversed map nodes to remain below 600, while the Theta* algorithm calculates that the the highest number of nodes is 3658. In terms of path length, the difference in the length of the paths planned by the two algorithms is not significant, and the paths planned by the Theta* algorithm are on average 1.60% shorter than those planned by the W-Theta* algorithm in the three groups of maps, and the Theta* algorithm has a slight advantage in the path length. Theta* algorithm is better than the W-Theta* algorithm in terms of spatial complexity. The memory consumed by W-Theta* algorithm is reduced by 15.02, 32.32, and 35.96% compared with Theta* algorithm in maps of $40 \times 40$ m, $80 \times 80$ m, and $120 \times 120$ m, respectively. It can be seen that the difference between the performance of W-Theta* algorithm and Theta* algorithm is more obvious in the larger size of the map environment.

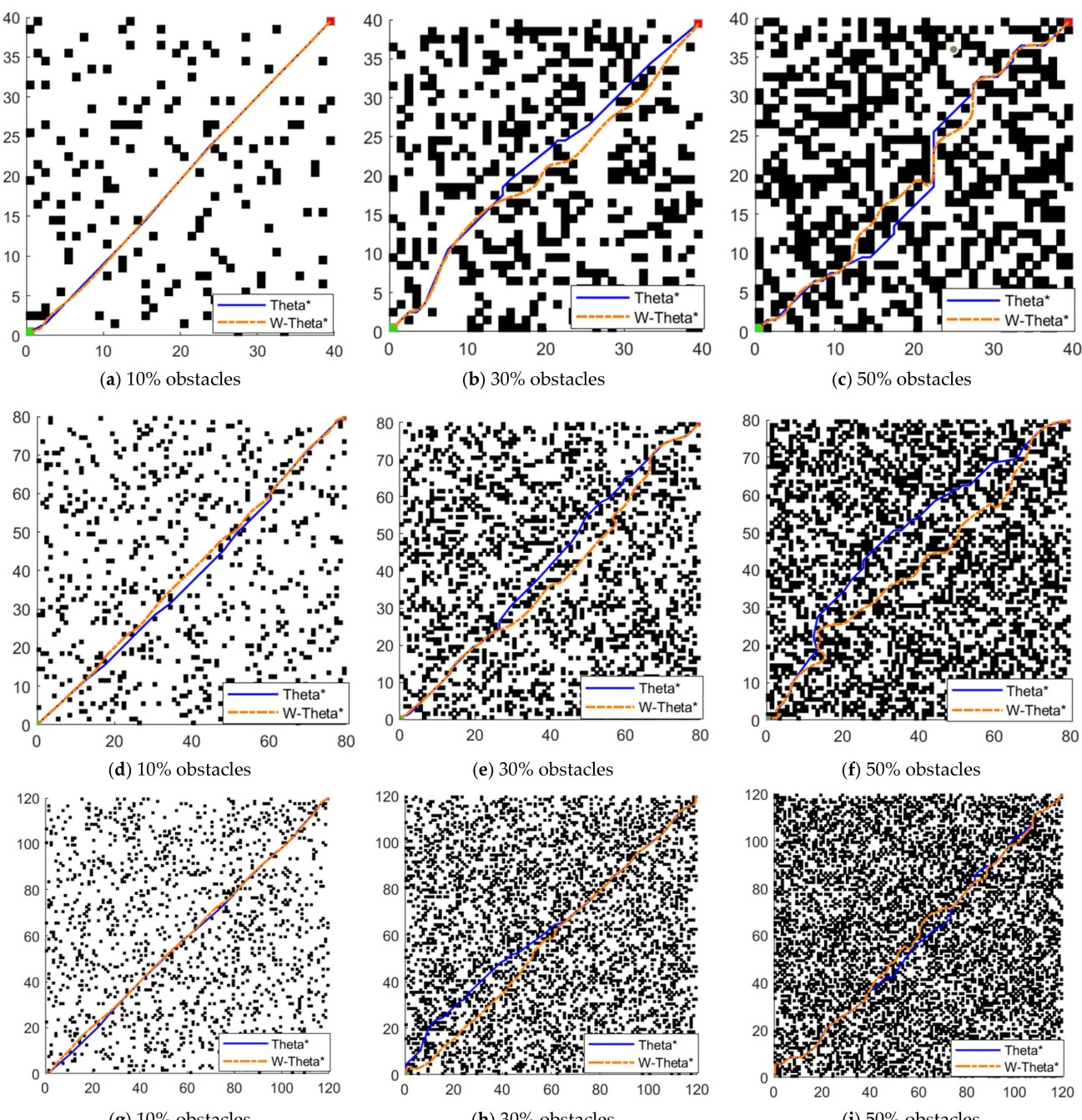

**Figure 5.** Comparison between W-Theta* algorithm and Theta* algorithm for maps with three sizes and different obstacle densities.

**Table 1.** Experimental data table of W-Theta* algorithm and Theta* algorithm for maps with three sizes and different obstacle densities.

| Map Size/m² | Obstacle Density | Algorithm | Pathfinding Time/s | Number of Traversal Nodes | Path Length/m | Memory Consumption/Bit |
|---|---|---|---|---|---|---|
| 40 × 40 | 10% | W-Theta* | 0.017162 | 181 | 56.2899 | 39,483 |
| | | Theta* | 0.030727 | 199 | 56.2899 | 40,971 |
| | 30% | W-Theta* | 0.013585 | 190 | 58.7589 | 55,331 |
| | | Theta* | 0.020874 | 314 | 57.7190 | 65,555 |
| | 50% | W-Theta* | 0.025756 | 205 | 63.0570 | 71,531 |
| | | Theta* | 0.068320 | 518 | 62.2417 | 96,451 |
| 80 × 80 | 10% | W-Theta* | 0.031979 | 378 | 114.4009 | 121,179 |
| | | Theta* | 0.229164 | 1045 | 115.6518 | 183,179 |
| | 30% | W-Theta* | 0.029807 | 343 | 118.3512 | 177,675 |
| | | Theta* | 0.199758 | 1344 | 115.8627 | 258,171 |
| | 50% | W-Theta* | 0.037211 | 356 | 130.7473 | 234,675 |
| | | Theta* | 0.221151 | 1731 | 123.3626 | 344,867 |
| 120 × 120 | 10% | W-Theta* | 0.054977 | 570 | 170.3327 | 246,419 |
| | | Theta* | 0.707187 | 2385 | 170.1267 | 391,355 |
| | 30% | W-Theta* | 0.053313 | 487 | 176.7601 | 371,787 |
| | | Theta* | 0.983099 | 3658 | 175.7588 | 625,867 |
| | 50% | W-Theta* | 0.060754 | 444 | 186.6482 | 492,251 |
| | | Theta* | 0.666634 | 3104 | 180.2422 | 705,835 |

### 3.2. Experiment of Comparing W-Theta* Algorithm with Other Algorithms

This part of the experiment uses the Moving AI lab public dataset on three types of maps to compare various path algorithms, the maps are Baldurs Gate II map, urban map and Warcraft III map, and the details of each map are shown in Table 2.

**Table 2.** Map related information table.

| Map Type | Map Size/m² | Map Name | Number of Obstacles | Max Length Problem in Scenario |
|---|---|---|---|---|
| Baldurs Gate II | 80 × 80 | AR0304SR | 1734 | 79.42640686 |
| | | AR0513SR | 2073 | 78.15432892 |
| | | AR0709SR | 2048 | 75.01219330 |
| City maps | 256 × 256 | Boston | 47,768 | 379.5290039 |
| | | Shanghai | 48,005 | 359.75945129 |
| | | Berlin | 47,540 | 363.33304443 |
| Warcaft III | 512 × 512 | Harvest moon | 114,594 | 567.93311615 |
| | | Scorched basin | 80,848 | 507.51175995 |
| | | Dusk wood | 127,229 | 615.9625534 |

The Dijkstra algorithm, A* algorithm, Theta* algorithm, and W-Theta* algorithm were applied on each map for comparative experiments in terms of pathfinding time, number of traversed nodes, path length, and memory usage, and the visualization results of their path planning are shown in Figure 6. The solid purple line in the figure is the path planned by the Dijkstra algorithm, the solid red line is the path planned by the A* algorithm, the solid blue line is the path planned by the Theta* algorithm, the dashed yellow line is the path planned by the W-Theta* algorithm, the green dot in each map indicates the starting point, the red dot indicates the end point. The white area is the passable area, and the other color grids represent different obstacles. The relevant data of this part of the experiment are shown in Tables 3–5. The performance of the proposed W-Theta* algorithm and other

mainstream algorithms is analyzed by comparing the path planning time, the number of grids in the traversed map, the length of the planned path, the computer resources consumed, and the turning angle of the path of each algorithm.

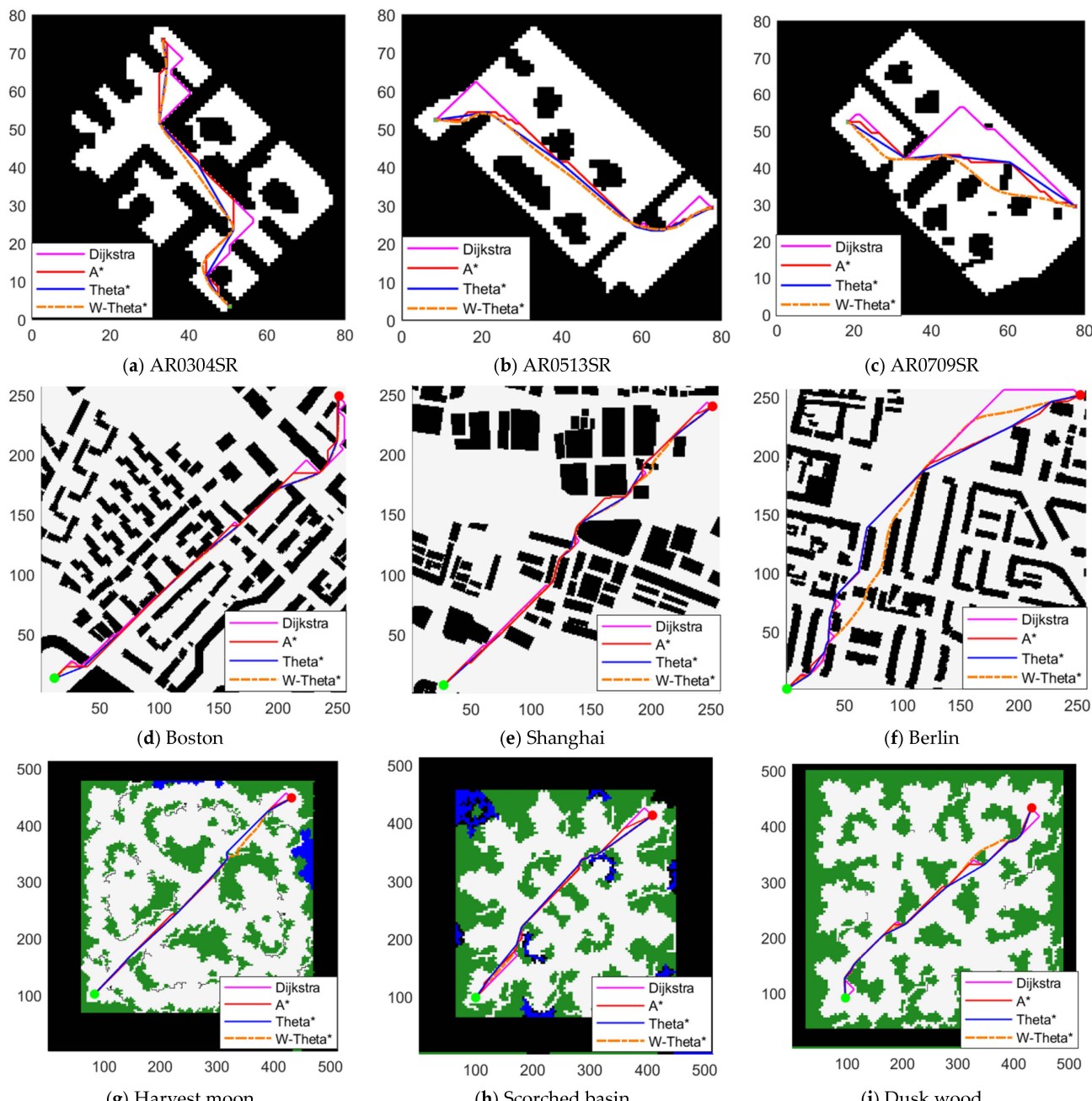

**Figure 6.** Comparative experimental results of various algorithms on public dataset maps.

**Table 3.** Experimental data sheet for various algorithms on Baldurs Gate II map.

| Map Name | Algorithm | Pathfinding Time/s | Number of Traversal Nodes | Path Length/m | Memory Consumption/bit | Total Path Turning Angle/° |
|---|---|---|---|---|---|---|
| AR0304SR | A* | 0.053776 | 611 | 85.4975 | 249,720 | 405.00 |
| | Dijkstra | 0.172302 | 1720 | 97.0955 | 252,208 | 450.00 |
| | Theta* | 0.065306 | 611 | 81.3935 | 335,867 | 140.95 |
| | W-Theta* | 0.033788 | 290 | 82.6155 | 222,792 | 140.95 |
| AR0513SR | A* | 0.199933 | 1473 | 86.1543 | 307,264 | 1170.00 |
| | Dijkstra | 0.217634 | 2067 | 96.0955 | 356,432 | 540.00 |
| | Theta* | 0.213901 | 1473 | 82.3920 | 387,939 | 183.73 |
| | W-Theta* | 0.060505 | 448 | 90.7401 | 223,744 | 121.00 |
| AR0709SR | A* | 0.077469 | 730 | 70.3553 | 248,408 | 1035.00 |
| | Dijkstra | 0.223962 | 2026 | 82.7817 | 353,456 | 270.00 |
| | Theta* | 0.087235 | 730 | 67.0026 | 329,115 | 198.43 |
| | W-Theta* | 0.030227 | 253 | 68.0747 | 210,064 | 131.55 |

**Table 4.** Experimental data sheet for various algorithms on City maps.

| Map Name | Algorithm | Pathfinding Time/s | Number of Traversal Nodes | Path Length/m | Memory Consumption/Bit | Total Path Turning Angle/° |
|---|---|---|---|---|---|---|
| Boston | A* | 15.296216 | 16,573 | 365.8721 | 2,425,264 | 1035.00 |
| | Dijkstra | 42.924916 | 47,688 | 393.2102 | 4,925,416 | 2250.00 |
| | Theta* | 15.504178 | 16,573 | 354.5103 | 2,792,580 | 280.83 |
| | W-Theta* | 0.892018 | 1576 | 358.4090 | 1,225,072 | 126.34 |
| Shanghai | A* | 8.034565 | 11,632 | 344.5290 | 2,026,280 | 2205.00 |
| | Dijkstra | 41.337722 | 47,792 | 354.4701 | 4,928,472 | 2160.00 |
| | Theta* | 8.226009 | 11,632 | 332.0240 | 2,390,588 | 488.93 |
| | W-Theta* | 0.535700 | 1260 | 333.6091 | 1,193,528 | 268.15 |
| Berlin | A* | 18.947835 | 22,984 | 401.0732 | 2,948,536 | 5985.00 |
| | Dijkstra | 42.740184 | 47,043 | 415.1564 | 4,882,776 | 2250.00 |
| | Theta* | 19.215597 | 22,984 | 381.4792 | 3,320,892 | 389.47 |
| | W-Theta* | 0.580014 | 1251 | 384.7610 | 1,205,720 | 266.51 |

**Table 5.** Experimental data sheet for various algorithms on Warcraft III maps.

| Map Name | Algorithm | Pathfinding Time/s | Number of Traversal Nodes | Path Length/m | Memory Consumption/Bit | Total Path Turning Angle/° |
|---|---|---|---|---|---|---|
| Harvest moon | A* | 49.332874 | 18,542 | 506.6194 | 8,333,752 | 2205.00 |
| | Dijkstra | 350.203546 | 111,916 | 515.7321 | 15,813,992 | 810.00 |
| | Theta* | 50.735335 | 18,542 | 498.0176 | 10,988,924 | 383.65 |
| | W-Theta* | 6.665620 | 1812 | 500.4910 | 6,990,296 | 223.22 |
| Scorched basin | A* | 74.824937 | 25,722 | 470.9382 | 9,983,080 | 3015.00 |
| | Dijkstra | 221.051927 | 79,414 | 490.8204 | 14,297,848 | 1350.00 |
| | Theta* | 76.241743 | 25,722 | 452.9727 | 13,177,868 | 466.23 |
| | W-Theta* | 8.908404 | 1684 | 456.8308 | 8,061,688 | 318.81 |
| Dusk wood | A* | 84.910705 | 31,517 | 521.4234 | 8,925,456 | 2250.00 |
| | Dijkstra | 373.709195 | 123,031 | 556.2174 | 16,260,704 | 1170.00 |
| | Theta* | 86.4172151 | 31,517 | 501.8894 | 11,362,259 | 329.87 |
| | W-Theta* | 8.642783 | 3077 | 509.5698 | 6,645,664 | 204.20 |

The data in Tables 3–5 gives the following information: the pathfinding time W-Theta* algorithm is the shortest, while the longest time consuming algorithm is Dijkstra algorithm. In Baldurs Gate II, W-Theta* reduces the pathfinding time by an average of 55.96% compared to the A* algorithm, 61.78% compared to the Theta* algorithm, and 79.70% compared to the Dijkstra algorithm; in City maps, W-Theta* reduces the pathfinding time by an average of 94.81%, 94.91% compared to Theta* and 98.24% compared to Dijkstra; in Warcaft III, W-Theta* reduced the pathfinding time by 88.13%, 88.39% compared to Theta* and 97.97% compared to Dijkstra. The number of nodes traversed by the W-Theta* algorithm is also the lowest, because Theta* is based on the A* algorithm, so both algorithms traverse the same number of nodes, and the highest number of nodes traversed by the Dijkstra algorithm, and the number of nodes traversed by the W-Theta* algorithm relative to the A* algorithm and the original Theta* algorithm is the highest. The nodes traversed by the W-Theta* algorithm are reduced by 62.49%, 91.41%, and 91.31% in Baldurs Gate II, City maps, and Warcaft III maps, respectively; compared to the original Theta* algorithm, the nodes traversed by the W-Theta* algorithm are reduced by 82.99, 97.13, and 97.13% in Baldurs Gate II, City maps, and Warcaft III maps, respectively, and 82.99%, 97.13%, and 97.92%, respectively. In terms of the length of the planned paths, the Theta* algorithm plans the shortest paths, but the path lengths planned by the W-theta* algorithm do not differ much from those planned by the Theta* algorithm, and the path lengths planned by the W-theta* algorithm relative to the Theta* algorithm increase by 1 in Baldurs Gate II, City maps, and Warcaft The path lengths planned by the W-Theta* algorithm compared to the Theta* algorithm increased by 1.98%, 0.81%, and 0.96% in Baldurs Gate II, City maps, and Warcaft III maps, respectively, which shows that the difference between the two algorithms is smaller in the larger environment; the path lengths planned by the W-Theta* algorithm compared to the A* algorithm decreased by 2.48 and 3.09%. The path lengths planned by the W-Theta* algorithm compared to the Dijkstra algorithm are reduced by 14.83, 7.35, and 6.09% for Baldurs Gate II, City maps, and Warcaft III maps, respectively. In terms of memory consumption, the least memory resources are used by the W-Theta* algorithm. Memory resources used by the W-Theta* algorithm compared to the A* algorithm are reduced by 17.80, 49.90, and 20.30% for the maps of Baldurs Gate II, City maps, and Warcaft III, respectively. Memory resources usage by the W-Theta* algorithm compared to the Compared to the Theta* algorithm is reduced by 37.39, 56.63, and 38.90% for Baldurs Gate II, City maps, and Warcaft III maps, respectively. The W-Theta* algorithm is compared to the Dijkstra algorithm, and memory resource usage is reduced by 17.80, 49.90, and 20.30% for Baldurs Gate II, City maps, and Warcaft III maps, respectively. Maps and Warcaft III by 29.82%, 75.41%, and 52.85%, respectively. In the three sets of maps, the paths planned by the W-Theta* algorithm have the smallest turning angles, and the paths planned by the A* algorithm have the largest turning angles. The W-Theta* algorithm reduces the turning angles by 87.06, 77.67, and 34.57% on average compared to the A* algorithm, Dijkstra algorithm, and Theta* algorithm.

From the results of this comparison experiment, we can conclude that the W-Theta* algorithm proposed in this paper is substantially optimized in terms of time complexity and space complexity compared with the A* algorithm, Dijkstra algorithm, and Theta* algorithm, and can achieve fast planning of high-quality paths in various complex environments. The W-Theta* algorithm plans paths with a smaller total turn angle and better overall path smoothing. In general, W-Theta* has significantly improved the efficiency of the algorithm while guaranteeing the resultant path, and the path smoothing process makes it more consistent with the motion of the robot in real situations.

## 4. Conclusions

To address the problem of slow path planning by traditional A* algorithm and Theta* algorithm in a large scene environment, a W-Theta* algorithm is proposed, and its generation of discrete path points is optimized for trajectory processing. The following conclusions can be drawn from two comparison experiments:

1. Compared with the traditional A* algorithm and the improved A*-based Theta* algorithm, the W-Theta* algorithm enables fast path planning in various complex environments by introducing a dynamic weighting strategy. Especially, the advantage is more obvious in the environment with larger map size, which solves the problem of slow path planning by traditional A* algorithm and Theta* algorithm in large scene environment. It is experimentally concluded that the W-Theta* algorithm reduces the path planning time by 81.65% on average compared with the Theta* algorithm, and by 79.59% on average compared with the A* algorithm.

2. To reduce unnecessary computations, the W-Theta* algorithm improves the algorithm performance by adding an additional value to the heuristic function to ensure uniqueness in planning the path, which allows it to control the traversal of nodes when planning the path and reduces the memory consumption during computation. According to the experimental data, it can be seen that the W-Theta* algorithm consumes an average of 44.31% less computer memory resources during computation than the Theta* algorithm, and an average of 29.33% less than the A* algorithm.

3. This is because the smaller the turn angle of the path is, the better the smoothing process is. In this paper, by using Euclidean distance for distance calculation, the total turn angle of the paths planned by the W-Theta* algorithm is reduced by 87.06, 77.67, and 34.57% on average with respect to the A* algorithm, Dijkstra algorithm, and Theta* algorithm. The paths planned by the W-Theta* algorithm are smoothed using the differential flattening method to fit the path points, which connects the sparse path points into smooth curves or dense trajectory points, ensuring the final paths are continuously smooth.

4. The paths planned by the W-Theta* algorithm are slightly longer than those of the original Theta* algorithm, so further optimization of the planned path length of the W-Theta* algorithm can be done.

**Author Contributions:** Conceptualization, Z.S.; methodology, Y.H.; software, Y.H. and J.L.; validation, Y.H., J.L. and Y.Z.; formal analysis, Y.Z.; investigation, Z.S.; resources, J.L.; data curation, Y.H.; writing—original draft preparation, Y.H.; writing—review and editing, Y.H.; visualization, J.L.; supervision, Y.Z.; project administration, Y.Z.; funding acquisition, Y.Z. All authors have read and agreed to the published version of the manuscript.

**Funding:** This research was funded by Research Project of China Disabled Persons' Federation—on assistive technology: 2022CDPFAT-01.

**Data Availability Statement:** Publicly available datasets were analyzed in this study. This data can be found here: https://movingai.com/.

**Conflicts of Interest:** The authors declare no conflict of interest.

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
