# Peer review of "Research on Path Planning of Mobile Robot Based on Improved Theta* Algorithm"

_algorithms, doi:10.3390/a15120477_

Round 1

Reviewer 1 Report

the quality of the paper can be improved by addressing the following questions:

1. Real world problems can be solved

2. Comparision with intelligent algorithms like PSO can be done

3. Computational complexity of the proposed algorithm is to described

4. Results and discussion is to be elaborated

5. Conclusion part is to be elaborated

Author Response

Dear Reviewers.

   Thank you very much for your valuable comments on this paper. We have taken all your comments into consideration and corrected them. The following section describes the corrections we have made to the article.

  1. We analyzed the time complexity and space complexity of the algorithm in our experiments by the algorithm running time and the memory consumption during computation, respectively.
  2. Based on your suggestions, we have enhanced the description and discussion in the experimental section.
  3. We have enhanced the conclusion of the experiment.
  4. The path planning we use is implemented based on discrete grid graph search techniques, which are fundamentally different for the techniques used in intelligent path planning algorithms (e.g., metaheuristic algorithm PSO, etc.). Since the so innovative elements proposed in this paper are implemented based on graph search techniques. Therefore, we believe that there is no significance with intelligent path planning algorithms for.
  5. We redefine the path smoothing metric by adding the turn angle of the path, reflecting the advantage of any angle path planning in path optimization.
  6. We re-describe the heuristic function h in detail, mainly reflected in Section 2.2 of this paper.
  7. We have redescribed the process steps of the Theta* algorithm to make it more detailed.
  8. We have corrected equation (3), the distance calculation formula we used in our experiments is Euclidean distance, but due to an oversight in writing, we did not correct it in time at that time, and we are very sorry. We have made the corrections and further described equation (4).
  9. We have further improved the readability of the article by re-explaining all the figures in the article through the use of legends.

Reviewer 2 Report

This paper proposes a novel path finding algorithm by adding multi-angle pathfinding idea and path smoothing technique. This reviewer believes it is very interesting and useful in the field of robotics. One suggestion is to the evaluation part: It will be useful to quantify the performance contribution of each of the multi-angle and smoothing to measure their relative significance.  

Author Response

(The authors gave the same response as above.)

Reviewer 3 Report

Title: “Research on path planning of mobile robot based on improved Theta* algorithm”

In this manuscript the authors proposed an improved Theta* algorithm, that is the W-Theta* algorithm. In particular, the heuristic functon of Theta* is improved by introducing a weighting strategy, while the default Euclidean distance calculation formula of Theta* is changed to a diagonal distance calculation formula, which finally achieves a reduction in computation time while ensuring a shorter global path. The trajectory optimization is achieved by curve fitting of the generated path points to make the motion trajectory of the mobile robot smoother. Simulation results show that the improved algorithm can quickly plan paths in large scenarios. Compared with other path planning algorithms, the algorithm has better performance in terms of time and computational cost. More specifically, the authors claim that in different scenarios, the W-Theta* algorithm reduces the computation time of path planning by 81.65% compared with the Theta* algorithm and 79.59% compared with the A* algorithm; the W-Theta* algorithm reduces the memory occupation during computation by 44.31% compared with the Theta* algorithm and 29.33% compared with the A* algorithm.

General comment: Although the topic of this work could be interesting, the quality of manuscript should be improved. In the following a short list of major and minor issues to be deeply reworked :

2. Algorithm design 133

2.1. Theta* algorithm

lines: "where( )f n is the estimated cost from the start point to the target point;( )g n is the actual cost from the start point to the current noden ; and( )h n is the heuristic function, which also indicates the estimated cost from the current noden to the target point."

*) These lines are not clear. Please explain better and provide explicitly the form of the h function.

Lines: “4. Extend noden . Iterate over all neighboring nodes of noden (set top ) and per- 181

form the following operations. 182

• If nodep is an obstacle or is already in the close table, do not process it, other- 183

wise proceed to the next step. 184

• Check whether the parent node of the current noden exists. If it exists, detect 185

whether nodep and node n are visible by line of sight (LOS) function [13], if 186

so, record the parent node of noden , otherwise record noden ; if not, record 187

noden . Name the final recorded node as nodek , and then proceed to the next 188

operation. 189

• If nodep is not in the open table, add it to the open table and set the parent 190

node of nodep to nodek . Finally, calculate the cost of nodep . If nodep is 191

already in the open table, we need to determine whether the cost required to 192

pass nodek is lower than the original cost, if yes, change the parent node of 193

nodep to node k and update the cost of nodep . If not, do not do any pro- 194

cessing.”

*) These lines are not clear. Please explain better for interested readers.

Lines: “1. Initialize the starting point and target point, create a new open table and a new close 172

table, and add the starting point to the open table. 173

2. Determine whether the open table is empty, if it is empty, the path planning fails and 174

the search stops; if it is not empty, the node with the lowest evaluation cost in the 175

open table is taken as the current node to be expanded and set ton . Go to the next 176

step. 177

3. Determine whether the current noden is the target point, if yes, it means the path 178

finding is successful, backtrack the parent node of n until the starting point is the 179

path, and the search is terminated; otherwise, go to the next step. 180

4. Extend noden . Iterate over all neighboring nodes of noden (set top ) and per- 181

form the following operations. 182

• If nodep is an obstacle or is already in the close table, do not process it, other- 183

wise proceed to the next step. 184

• Check whether the parent node of the current noden exists. If it exists, detect 185

whether nodep and node n are visible by line of sight (LOS) function [13], if 186

so, record the parent node of noden , otherwise record noden ; if not, record 187

noden . Name the final recorded node as nodek , and then proceed to the next 188

operation. 189

• If nodep is not in the open table, add it to the open table and set the parent 190

node of nodep to nodek . Finally, calculate the cost of nodep . If nodep is 191

already in the open table, we need to determine whether the cost required to 192

pass nodek is lower than the original cost, if yes, change the parent node of 193

nodep to node k and update the cost of nodep . If not, do not do any pro- 194

cessing. 195

5. Remove the already traversed node n from the open table and add it to the close 196

table, then return to step 2.

The heuristic function of the improved algorithm is mainly to find the shortest path. 213

Using the traditional method of expanding nodes will generate many useless expansion 214

nodes and increase the computational cost. To address this problem, the following im- 215

provements are made in this paper: 216

1. Use the diagonal distance to express the distance between the current node and the 217

end point, which can be expressed as: 218( ) ( 2 2) min( , )x y x ydist n d d d d= + + −

(3)

wherexd isn goalx x− andyd isn goaly y− , wheren( , )nx y is the coordinates of 219

the current noden and( , )goal goalx y is the coordinates of the target point.( )dist n 220

is the diagonal distance of the current noden . 221

lines:” 2. In this paper, we will add additional values to the heuristic function( )h n , the size 222

of which is the vector fork product of the initial point-to-target vector and the current 223

point-to-target vector, and then change the value of( )h n to make it more inclined 224

to the connection from the initial point to the target point in selecting the path, to 225

ensure uniqueness in planning the path and reduce unnecessary computation. The 226

functional expression is: 227_ _c=V Vc e s e

(4)

where c denotes the vector fork product of the start-point-to-target-point vector_Vs e 228

and the current node-to-target-point vector_Vc e . 229

3. By adding a weightw to the heuristic function( )h n and then dynamically adjust- 230

ing this weight according to the progress of the algorithm, the importance of the heu- 231

ristic function is reduced by decreasing the weight as the path planned by the algo- 232

rithm approaches the target point, while increasing the relative importance of the 233

true cost of the path.

*) Equations (3) and (4) are not totally clear. Please rework and improve.

  1. Simulation and 4 .Conclusions

*) This work does not have the standard sections of a scientific contribution. In particular, it lacks of the Results section (in this case perhaps “Simulations”) and of Discussion section, where the authors compare their main achievements to the current state of the art. Please rework all these part according to this comment.

*) All tables and figures should have more descriptive captions, please rework.

Author Response

(The authors gave the same response as above.)

Round 2

Reviewer 1 Report

I appreciate your contribution for this paper

Reviewer 3 Report

Title: “Research on path planning of mobile robot based on improved Theta* algorithm”

In this manuscript the authors proposed an improved Theta* algorithm, that is the W-Theta* algorithm. In particular, the heuristic functon of Theta* is improved by introducing a weighting strategy, while the default Euclidean distance calculation formula of Theta* is changed to a diagonal distance calculation formula, which finally achieves a reduction in computation time while ensuring a shorter global path. The trajectory optimization is achieved by curve fitting of the generated path points to make the motion trajectory of the mobile robot smoother. Simulation results show that the improved algorithm can quickly plan paths in large scenarios. Compared with other path planning algorithms, the algorithm has better performance in terms of time and computational cost. More specifically, the authors claim that in different scenarios, the W-Theta* algorithm reduces the computation time of path planning by 81.65% compared with the Theta* algorithm and 79.59% compared with the A* algorithm; the W-Theta* algorithm reduces the memory occupation during computation by 44.31% compared with the Theta* algorithm and 29.33% compared with the A* algorithm.

General comment: Although the authors revised this manuscript some issues should be still reworked: In particular, I recommend a better organization of the main text following the standard section of a scientific contribution, an improvement of the figure captions and the improvement of the language to allow the interested readers to better follow the logic flow of the work.